# Rice Bran as an Alternative Feedstuff in Broiler Nutrition and Impact of Liposorb^®^ and Vitamin E-Se on Sustainability of Performance, Carcass Traits, Blood Biochemistry, and Antioxidant Indices

**DOI:** 10.3390/vetsci10040299

**Published:** 2023-04-17

**Authors:** Youssef A. Attia, Elwy A. Ashour, Sameer A. Nagadi, Mayada R. Farag, Fulvia Bovera, Mahmoud Alagawany

**Affiliations:** 1Agriculture Department, Faculty of Environmental Science, King Abdulaziz University, Jeddah 21589, Saudi Arabia; 2Poultry Department, Faculty of Agriculture, Zagazig University, Zagazig 44511, Egypt; 3Forensic Medicine and Toxicology Department, Faculty of Veterinary Medicine, Zagazig University, Zagazig 44511, Egypt; 4Department of Veterinary Medicine and Animal Production, University of Napoli Federico II, via F. Delpino, 1, 80137 Napoli, Italy

**Keywords:** rice bran, vitamin E-Se, Liposorb^®^, growth, carcass, metabolic markers, antioxidants, broilers

## Abstract

**Simple Summary:**

To find alternatives to common ingredients, possibly of local origin, is very important considering the most recent global crisis. Rice bran could be a valuable candidate for poultry diets, but its chemical–nutritional characteristics and the effects on living birds must be carefully evaluated to identify all the possible negative or positive effects. In addition, the use of supplements in combination with rice bran in poultry diets can be a solution to recover some negative effects of this ingredient.

**Abstract:**

The impact of dietary rice bran with or without feed additives on the performance, carcasses, and blood profiles of chickens was examined. A total of 245 unsexed one-week-old broiler chicks were divided into seven groups, with seven replications of five chicks each. The treatments were: (1) control, (2) 5% rice bran, (3) 5% rice bran + 0.5 g/kg of Liposorb^®^, (4) 5% rice bran + 1 g/kg of vitamin E-Se, (5) 10% rice bran, (6) 10% rice bran + 0.5 g/kg Liposorb^®^, and (7) 10% rice bran + 1 g/kg of vitamin E–selenium. Considering the entire experimental period, it did not affect the in vivo performance of the broilers. However, all the experimental diets decreased dressing % compared with the control (*p* < 0.01) and the worst values were obtained for the 10% RB groups (75.7, 75.9, and 75.8%, respectively, for 10%RB, 10%RB + Liposorb, and 10%RB + Vit. E-Se groups). All the experimental diets decreased (*p* < 0.01) the albumin/globulin ratio due to an increased level of serum globulins. Differences in lipid profiles, antioxidants, and immunity parameters in plasma were not related to dietary treatments. In conclusion, the use of rice bran up to 10% in diets had no harmful effect on the overall growth performance of the broilers from 1 to 5 weeks of age. Still, carcass characteristics were negatively affected, except for heart percentage. In addition, the supplementation of Liposorb^®^ or vitamin E-Se to rice bran diets did not recover these harmful effects. Thus, rice bran could be utilized at 10% in broiler diets when growth performance was considered; further research is required.

## 1. Introduction

Alternative feed resources and agricultural by-products have become vital solutions for farm animal nutrition during the COVID-19 crisis and the recent Russia–Ukraine war. This occurred because of the lack of dry boards, seaports, and airports and the complicated international relations that restricted imported feed resources for livestock feeding in developing countries [1,2]. Local agricultural by-products can be a good solution; rice bran is of great interest for poultry feeding due to its considerable nutrient content [3,4]. However, several factors limit rice bran utilization in poultry nutrition, e.g., its low protein and high fiber and fat content and the presence of antinutritional agents such as phytic acid. The use of rice bran in broiler diets of up to 7.5% decreased performance [5]. Some methods and techniques such as fermentation, supplementation of enzymes, and use of the fermented product can improve rice bran’s quality and thus improve the performance of layer chickens and broiler chickens when they are fed diets containing rice bran [3,6,7].

In the current study, supplementations of emulsifiers and vitamin E-Se were used in an attempt to increase the rice bran quality. Emulsifiers are used in poultry diets for enhancing productive performance and digestion coefficients of the nutrients, especially lipids, and include milk-derived casein, soy lecithin, lysolecithin (lecithin) or lysophatidylcholine, sodium stearoyl-2-lactylate, glycerol polyethylene glycol ricinoleate, and bile salt [8,9]. The emulsifiers can be used in poultry diets in combination with different feed ingredients and different levels of metabolizable energy to enhance digestion and/or absorption. The use of soy lecithin in broiler chicken diets can change critical events in the expression of hepatic genes linked to fat metabolism alterations. The modes of action of lecithin derived from plant oils have potential synergistic activities with α-tocopherol to stabilize polyunsaturated fatty acids in lipid oxidation studies [7]. Cantor et al. [10] used soybean lecithin instead of vegetable–animal fat in broiler diets.

Vitamin E and selenium play essential roles in different bodily functions, one of which is an intercellular antioxidant effect [11,12]. Vitamin E prevents the oxidation of polyunsaturated fatty acids in the cells, protecting the cell membrane from oxidative impairment caused by reactive oxygen species [13,14]. In addition, the use of vitamin E in the poultry diet improves growth performance. Abou-Kassem et al. [13] found that a dietary addition of 250 mg of Vit. E/kg ration increased live body weight at six weeks of age, feed efficiency, and body weight gain from 1 to 6 weeks of age in comparison with the control. However, studies investigating the influence of dietary rice bran with Liposorb^®^ and Vitamin E-Se emulsifier on chickens’ performance, carcass, and blood chemistry are not found. Our research aimed to verify whether feed additives such as Liposorb^®^ and vitamin E-Se emulsifier are able to improve the rice bran quality and utilization by broiler chickens by feeding them with diets containing rice bran with and without supplements to assess the effects of different diets on growth performance, carcass traits, antioxidants and immunity status, and liver and kidney functions.

## 2. Materials and Methods

### 2.1. Experimental Design and Diets

The Deanship of Scientific Research (DSR), King Abdulaziz University (KAU), Saudi Arabia approved this research work under protocol no. IFPIP 268-155-1443. The protocol recommended general humane treatment of animals that did not cause pain, suffering, distress, or lasting harm to animals according to the Royal Decree number M59 in 14/9/1431H and institutional approval code ACUC-22-1-2.

The study was carried out at Zagazig University, Zagazig, Egypt, in conjunction with King Abdulaziz University, Jeddah, Saudi Arabia, to investigate dietary rice bran with or without feed additives for feeding broiler chickens from 1 to 35 days old. A total of 245 unsexed one-week-old broiler (Arbor Acres) chicks were purchased from Dakahlia Poultry, Nasar City, Cario, Egypt. The chicks were housed in battery brooders (50 × 100 × 40 cm each) furnished with rice husks and provided with a 23:1 light–dark cycle from 1 to 7 days old and 20:4 from 8 d to the end of the experiment. The chickens were haphazardly divided into 7 groups with 7 replications of 5 chicks each. The duration of the trial was 4 weeks. All the groups were fed isocaloric and isonitrogenous diets and were organized as follows: (1) control, (2) 5% rice bran, (3) 5% rice bran + 0.5 g/kg of Liposorb^®^, (4) 5% rice bran + 1 g/kg of vitamin E-Se, (5) 10% rice bran, (6) 10% rice bran + 0.5 g/kg Liposorb^®^, and (7) 10%, rice bran + 1 g/kg of vitamin E-Se. The additives were added at the top of the diet, replacing the same amount of feed mixture. The levels of the additives were chosen based on previous research [3,6,7,8]. The chemical composition (Table 1) of rice bran was determined [15].

Based on the Arbor Acres’ broiler guidelines (https://www.thepoultrysite.com/focus/aviagen/aviagen-arbor-acres, accessed on 3 October 2022), the diets (Table 2) were formulated to meet broiler requirements [15]. Liposorb^®^ emulsifier was bought (Ceva Polchem Private Limited, Shivajinagar, Pune 411016, India), each 1 kg contained: 500 g lecithin, 100 g polyethyl glycerol ricinoleate, 2.16 g sodium chloride, 9.0 g silicon dioxide, and 388.84 calcium carbonate. Vitamin E-Se was bought from Multi Vita Company, 6th of October City, Egypt. Chemical nutritional characteristics of the diets were determined [15] or calculated [16].

### 2.2. Growth and Carcass Measurements

Chicks’ live weights and feed consumption were recorded at 1, 3, and 5 weeks of age to calculate the weight gain and feed conversion ratio on a replicate basis. The European production efficiency index was calculated according to the Arbor Acres’ breeder guide (https://www.thepoultrysite.com/focus/aviagen/aviagen-arbor-acres, (accessed on 3 October 2022), using the formula: (Average grams gained/day X% survival rate)/Feed Conversion X 10.

At the end of the trial, 7 birds from each treatment were selected to represent all treatment replicates and the average body weight of each treatment. Broilers were slaughtered according to the Islamic method (Halal) after being fasted for 8 h for carcass examination (carcass yield, dressing, and inner organs percentages).

### 2.3. Blood Constituents’ Assay

At the end of trial, the collection of blood samples (*n* = 7 per treatment, representing all treatment replicates) was accomplished in heparinized tubes. Then, they were centrifuged at an rpm of 2146× *g* for 15 min to separate the plasma sample. Plasma metabolites, including protein and its fraction, aspartate aminotransferase (AST), alanine aminotransferase (ALT), lactate dehydrogenase (LDH), creatinine, urea, triglycerides (TG), total cholesterol (TC) and its fractions (high-density lipoprotein-HDL, low-density lipoprotein-LDL, and very low-density lipoprotein-VLDL), and immunoglobulins (IgG, IgA, and IgM) were determined using an automatic analyzer with a commercial kit from Bio-diagnostic Company (Giza, Egypt), according to the manufacturing procedure. Plasma lysosomal activity was assessed with a 96-well microplate turbidity assay. Malondialdehyde (MDA), superoxide dismutase (SOD), catalase (CAT), total antioxidant capacity (TAC), and reduced glutathione (GSH) were measured colorimetrically using a microplate spectrophotometer with a commercial detection kit (Bio-diagnostic©, El Tahrir, Ad Doqi, El Omraniya, Giza Governorate, Egypt), following the manufacturer’s instructions.

### 2.4. Statistic Analysis

The normality of the data and the error distribution were evaluated using the Shapiro–Wilk test [16]. The random selection of the broilers and samples assured the respect of the 4 assumptions of analysis of variance (ANOVA). The homoscedasticity (variance homogeneity) was evaluated by Levene’s test SAS^®^ [17].

Data were analyzed with the SAS^®^ [15] software package using a two-way ANOVA (GLM procedure) according to the following model: Yij = m + RBi + Aj + RB×Aij + eijk, where Y is a single observation, m is the general mean, RB is the effect of the rice bran, A is the effect of the other additives (Liposorb and Vit. E), and e is the error. The replicate was the experimental unit. Mean comparison took place using the orthogonal contrast analysis [17], considering significant *p*-values lower than 0.05. In particular, the tested contrasts were: (1) control vs. all 5% RB bran groups regardless of supplementation; (2) control vs. all 10% RB bran groups regardless of supplementation; (3) 5 groups vs. 10% RB bran groups regardless of supplementation; (4) 5 vs. 10% RB unsupplemented (NS) groups; (5) control vs. 5% RB NS; (6) control vs. 10% RB NS groups; (7) Liposorb vs. Vit. E-Se groups regardless of RB level; (8) 5% RB NS vs. 5% RB + Liposorb; (9) 5% RB NS vs. 5% RB + Vit. E-Se; (10) 10% RB NS vs. 10% RB + Liposorb; (11) 10% RB NS vs. 10% RB + Vit. E-Se.

## 3. Results

To simplify the tables, when a contrast was not significant for all the studied criteria reported in a table it was not indicated.

### 3.1. Growth Parameters

Based on the data in Table 3, RB decreased BWG of birds (*p* < 0.01) in the first 3 weeks of the trial, while an opposite effect was shown in the following period (*p* < 0.01); thus, considering the entire period of the trial, BWG was not different among groups supplemented with different levels of RB. FI was affected (*p* < 0.01) only in the first period of the trial, with a negative effect due to increasing levels of RB. FCR was improved in 10% RB groups compared with the control (*p* < 0.01) from 3 to 5 weeks of age. Liposorb gave better (*p* < 0.01) BWG between 1 and 3 weeks and lower (*p* < 0.01) from 3 to 5 weeks compared with vitamin E, while vitamin E improved (*p* < 0.01) FI from 3 to 5 weeks of age and the EPEI value in comparison with the Liposorb. The contrast analysis showed that, considering the whole period of the trial, the control group had the best effects in terms of BWG and FI, while no differences were compared with the other groups. Regarding the FCR between 1 and 5 weeks of the trial, only the group supplemented with 10% RB + Vit. E-Se showed worsened values (*p* < 0.05) when compared with 10% RB unsupplemented group.

### 3.2. Carcass Measurements

The inclusion of increasing levels of RB in broiler diets reduced (*p* < 0.01) in a stepwise pattern almost all the carcass traits reported in Table 4 (except the heart %) compared with the control. The additives had no effect on carcass traits except for the gizzard and giblet %, which was higher (*p* < 0.01) when the groups were supplemented with Liposorb in comparison with the vitamin E and selenium. In general, the control group showed the highest values for all the criteria reported in Table 4 and the values are higher (*p* < 0.01) in the 5% groups compared with the 10% groups.

The use of Liposorb and vitamin E increased (*p* < 0.01) carcass and dressing % in the 5% RB groups, but the same did not occur in the 10% RB groups, in which the only supplementation with vitamin E and selenium decreased the liver % in comparison with the unsupplemented group.

### 3.3. Blood Parameters

Total protein (TP) and globulin (Glb) levels increased in a stepwise pattern (*p* < 0.01) in birds that were fed increasing levels of RB compared with the control group, while an opposite trend showed the albumin (Alb) to globulin ratio (Table 5). The albumin concentration in the blood of the 10% RB groups was lower (*p* < 0.01) compared with the control and the 5% RB groups. The supplementation of Liposorb improved TP, Alb, and Alb/Glb, while decreased Glb (*p* < 0.01) compared with the Vit. E-Se groups.

According to the contrast analysis, in the 5% RB groups, the supplementation of Liposorb and Vit. E-Se increased (*p* < 0.01) TP, while Vit. E + Se supplements decreased TP in the 10% RB groups; in both cases Vit. E-Se was the most potent supplement. The AST level in the 5% RB group was reduced (*p* < 0.05) due to the use of Liposorb. However, Liposorb increased the urea/creatinine ratio in the 5% RB groups compared with the Vit. E-Se, while in the 10% RB groups the Liposorb decreased (*p* < 0.01) the urea/creatinine ratio compared with the Vit. E-Se.

### 3.4. Lipid Profiles

The data on the lipid profile of broilers according to the experimental groups are reported in Table 6. The inclusion of RB at 5% decreased tryglicerides and vLDL levels compared with the groups that were fed 10% RB (*p* < 0.01). The use of Vit. E-Se increased HDL (*p* < 0.01) compared with Liposorb.

According to the contrast analyses, Liposorb showed several effects when broilers were fed diets containing 10% of RB; in fact, it reduced cholesterol, LDL, and LDL/HDL while increasing triglycerides and vLDL compared with the unsupplemented 10% RB group. Vit. E-Se also increased triglycerides and vLDL compared with the unsupplemented 10% RB group (*p* < 0.01). Triglycerides and vLDL levels were lower in the 5% groups than in the 10% RB groups (*p* < 0.01).

No effects could be observed for the antioxidant and immunological indices reported in Table 7, according to the inclusion of RB and/or supplements. The contrast analysis showed that the 5% RB groups had lower levels of IgM (*p* < 0.05) compared with the 10% RB groups.

## 4. Discussion

The reduction in growth rate due to RB inclusion in the first period of the trial (1–3 weeks of age) could be due to the presence of antinutritional factors affecting the digestive process in young birds. In fact, RB contains high levels of non-starch polysaccharides (NSPs), along with phytase (12.8 g/kg vs. 2.0 g/kg in maize), which are not digested by young birds and can increase digesta viscosity, thus reducing nutrient digestibility in the small intestine of chicken [15,18]. Studies have shown that the high content of NSP in broiler diets negatively affects growth performance [19]. However, the negative effects of RB diets were recovered in the last weeks of the trial in such a way that, considering the entire period, no differences could be detected between groups. The body gain between 3 and 5 weeks of age improved with 10% RB with or without vitamin E-Se compared with the other groups, indicating an increased tolerance of broilers to the RB diet with increasing animal age and thus gastrointestinal maturity. This is in partial agreement with Ahmad et al. [20], who found that broilers that were fed fermented rice bran diets had a significant difference in body weight gain with the control group through the first 2 weeks of age. This result could be attributed to the high metabolizable energy level of full-fat rice bran [20]. However, the high-fat content in rice bran would have decreased the palatability of the diets, thus reducing the feed consumption during the starter period [21]. Another possible explanation is that NSP can have a positive impact on broiler gastrointestinal health and performance, namely through its role as a source of substrates for probiotic bacteria species [22]; therefore, our hypothesis is that, over time, there may be a prevalence of prebiotic effects over those reducing digestibility and thus the health condition of the gastrointestinal system of chickens may, in general, improve. The potent effect of Liposorb on the growth rate of broilers between 1 and 3 weeks of age, which disappeared thereafter, indicated that increased lipid digestibility confirmed our hypothesis.

The broilers that were fed 10% RB without any additive recorded the best FCR between 3 and 5 weeks of age. The improvement in the feed conversion ratio in the 10% rice bran diet could possibly be due to better nutrient absorption [23]. The use of 2.5% or 5% lecithin in the broiler diets did not affect animal performance [10]. Birds that were fed diets enriched with 0.1% lecithin had better daily gain and feed intake than the control, while FCR was improved compared with 0.05% lecithin and the control. The use of 0.1% lecithin showed the highest relative liver weight compared with 0.05% lecithin and the control group at 21 and 42 days old [24].

Broilers in the control group recorded the maximum intake of the finisher feed. In contrast, birds that were fed fermented rice bran had better FCR than the other groups, while the fermented rice bran diets had a non-significant effect on FCR during the first 2 weeks [20]. In addition, during the starter period (1–8 days old), the feed intake of broilers receiving a corn–soybean diet was significantly higher than those receiving 20 and 30% rice bran diets [25].

The output of the current experiment demonstrates that using rice bran with vitamin E-Se or emulsifiers (Liposorb^®^) does not have any positive effects on performance. Similarly, Mujahid et al. [26] did not find any significant effect on the growth of broilers when studying the impact of antioxidants up to 250 ppm in the rice bran diet. Furthermore, birds consuming diets with 6.5% of rice bran stabilized by physical treatments (heat) showed an improved growth rate and the use of synthetic or natural antioxidants was not required [27]. On the other hand, Sun et al. [25] observed that 18% rice bran (full-fat) significantly improved the body weight gain of geese between 57 and 70 days old. Adrizal et al. [28] reported that diets with 0%, 7.5%, 15%, and 22.5% defatted rice bran did not affect chickens’ body weight or FCR.

Birds that were fed rice bran fermented by *Aspergillus flavus* showed a significant difference in weight gain vs. other groups during the 3rd week of age [20]. Our results can be supported by the outputs of Zare-Sheibani et al. [29], who reported that broiler chickens on rice bran (20 and 30%) diets had lower body weight gains and FCR compared with those feeding on extruded rice bran (20 and 30%) diets. In line with our results, Gallinger et al. [30] observed a reduction (−3.6 and −8.0%) in body weight gains of birds that were fed 30 and 40% rice bran, respectively. In geese, the birds that were fed a diet containing 30% and 40% defatted rice bran had a lower final body weight and body gain than the control group [31]. In addition, the feed consumption and growth rate declined in ducks when rice bran inclusion increased from 0 to 600 g/kg and adding enzymes to rice bran diets improved growth rate but not feed consumption and FCR [32]. The improvement in growth rate with enzyme supplementation may be due to enhanced dry matter digestibility [23].

Our data are confirmed by those obtained by Anitha et al. [33], who found that the carcass parameters viz. weights of carcass and giblets did not significantly differ among the treatments by crude rice bran oil levels. Rice bran oil did not affect giblet weight and eviscerated carcass weight of broilers. Similar results were reported by Purushothaman et al. [34]. Rice bran, phytase, and their interaction did not affect broilers’ gizzard and liver weight [35]. On the contrary, El-Ghamry et al. [21] reported that increasing the level of rice bran increased the gizzard weight. Increasing the gizzard weight could be attributed to the high crude fiber content in rice bran, which induced gizzard enlargement and increased gizzard weight.

Very few data are available in the literature on the effect of RB inclusion in the diet, with or without other supplements, on the blood protein profile of broilers. Our results partly agree with Fafiolu et al. [36], who observed that increasing levels of RB in broiler diets (from 10 to 30%) progressively increased the total protein concentration in the blood; however, the same authors found no effect of RB inclusion on the levels of albumin and globulin. In our trial, a very interesting result was represented by the increase of globulin due to an increase in the RB level, but also to the supplementation with Liposorb and Vit. E + Se, which was mainly responsible for the lower value of Alb/Glob in the experimental groups compared with the control. This result suggested a higher immune competence of the experimental groups compared with the control. Chen et al. [31] observed no effects of RB up to 40% on the blood profile of geese. According to Kang and Kim [37], ALT, AST, glucose, and albumin levels were also unaffected by rice bran treatments.

According to several authors [32,38,39], in our trial no effects of RB and/or supplements were detected on lipid parameters, including the total cholesterol and triglycerides. On the contrary, Crespo and Esteve-Garcia [40] stated that the total cholesterol content was significantly lowered in birds that received rice bran oil compared with other groups that were fed palm and tallow oil. Anitha et al. [41] found that rice bran extracts in broiler diets decreased total plasma cholesterol. In broilers, rice bran can become rancid and can decrease the lipid stability of meat and animal growth [42]. Therefore, cholesterol-lowering effects are present in rice bran oil [43]. The reduction of cholesterol by increasing the levels of rice bran in the diets could be related to the high level of fatty acids and/or crude fiber in rice bran, which increases soap compounds and insoluble complexes and leads to the lower absorption of cholesterol. Similarly, cholesterol-lowering influences of rice bran were approved by El-Ghamry et al. [21] and Attia et al. [44].

The inclusion of rice bran in diets did not affect levels of IgM but linearly improved the concentrations of IgG, according to the data obtained by Kang and Kim [37]. IgG is the dominant type of antibody found in blood and extracellular fluid, allowing it to control infections of body tissues by binding many kinds of pathogens such as viruses, bacteria, and fungi. In addition, the supplementation of fermented rice bran extract did not affect IgA and IgG levels in broiler chickens [45]. Furthermore, Raju et al. [38] found that sheep red blood cell response plasma levels were not affected by rice bran oil lysolecithin. However, the lack of adequate results on the role of rice bran and its derivatives in changing blood indices in poultry requires more research.

On the other hand, soy lecithin was suitable for enhancing the antioxidant system [32]. Dalia et al. [46] reported that the synergistic effect of Se and Vit. E was clear in the plasma level of IgM at day 42 and in the expression of splenic cytokines (IFN-γ, TNF-α, IL-2, and IL-10). The use of 100 mg/kg Vit. E showed the highest IgM level. The main impact of Se was that the different sources improved Ig A and G levels compared with the control group at 42 days of age.

## 5. Conclusions

From these results, rice bran up to 10% in diets did not harm the growth, feed intake, FCR, or production index of broiler chickens between 1 and 5 weeks of age, but dressing percentage was decreased and Liposorb^®^ (0.5 g/kg diet) and/or vitamin E-Se did not recover this trait. There were no significant differences regarding lipid profile, antioxidants, and immunity parameters in plasma. Thus, rice bran could be utilized at 10% in broiler diets, however further research is required to solve the problem of its negative effects on dressing percentage.

## Figures and Tables

**Table 1 vetsci-10-00299-t001:** Nutrient composition of rice bran.

Nutrients *	%
DM	89.77
OM	93.46
Ash	6.54
CP	14.89
CF	7.88
EE	17.32
NFE *	53.37

* DM—dry matter, OM—organic matter, CP—crude protein, CF—crude fiber, EE—ether extract, NFE (nitrogen free extract = 100 − (14.89 + 7.88 + 17.32 + 6.54).

**Table 2 vetsci-10-00299-t002:** Composition of the experimental (starter 8–21 d) and grower (22–35 d) diets.

Item	Starter Diets			Grower Diets		
	0 RB	5 RB	10 RB		5	10
Ingredient (%)						
Maize 8.5%	53.03	48.73	44.0	59.26	54.70	50.10
Soybean meal 44%	35.00	34.35	33.8	27.0	26.38	25.80
Corn gluten meal 62%	5.00	5.00	5.00	5.00	5.00	5.00
Rice bran	0.00	5.00	10.00	0.0	5.00	10.0
Soybean oil	2.90	3.00	3.25	4.82	5.01	5.20
Limestone	1.40	1.38	1.38	1.37	1.37	1.40
Di-calcium phosphate	1.50	1.45	1.44	1.55	1.54	1.50
NaCl	0.30	0.30	0.30	0.30	0.30	0.30
Premix ^1^	0.30	0.30	0.30	0.30	0.3	0.30
L-Lysine	0.15	0.13	0.12	0.15	0.15	0.15
Dl-Methionine	0.12	0.11	0.11	-	-	-
Choline chloride (50%)	0.30	0.25	0.30	0.25	0.25	0.25
			Chemical–nutritional characteristics, %			
ME, Kcal/Kg^2^	3025	3024	3024	3221	3223	3222
Fat ^3^	5.32	5.91	6.62	7.41	8.11	8.75
CP ^3^	23.01	23.00	23.00	20.2	20.0	20.0
Crude fiber ^3^	3.54	4.15	4.62	3.33	3.82	4.27
Ca ^2^	1.02	1.00	1.00	1.00	1.00	1.00
Nonphytate P ^2^	0.45	0.45	0.45	0.45	0.45	0.45
Lysine ^2^	1.32	1.30	1.30	1.10	1.10	1.10
Total sulfur aminoacids ^2^	0.92	0.92	0.92	0.72	0.72	0.72
DM ^3^	88.06	88.29	88.23	86.40	86.43	86.45
Ash ^3^	6.60	6.95	7.31	6.11	6.46	6.81
NDF ^3^	10.35	11.02	11.64	10.27	10.91	11.55
ADF ^3^	3.67	4.06	4.45	3.40	3.78	4.17
Lignin ^3^	0.47	0.65	0.82	0.48	0.65	0.83
Starch ^3^	42.01	40.25	38.19	46.08	44.13	42.16

^1^ Provides per kg of diet: vitamin D3, 5000 IU; vitamin A, 12,000 IU; vitamin K3, 3.605 mg; vitamin E, 130.0 mg; vitamin B1 (thiamin), 3.0 mg; vitamin B6, 4.950 mg; vitamin B2 (riboflavin), 8.0 mg; vitamin B12, 17.0 mg; D-biotin, 200.0 mg; folic acid, 2.083 mg; niacin, 60.0 mg; calcium D-pantothenate, 18.333 mg; iron, 80.0 mg; manganese, 100.0 mg; zinc, 80.0 mg; iodine, 2.0 mg; copper, 8.0 mg; selenium, 150.0 mg; cobalt, 500.0 mg. ^2^ Calculated values. ^3^ Determined values.

**Table 3 vetsci-10-00299-t003:** Growth performance of broiler chickens as affected by dietary rice bran treatments during the experiment.

	BWG, 1–3 w, g	BWG, 3–5 w, g	BWG, 1–5 w, g	FI, 1–3 w, g	FI, 3–5 w, g	FI, 1–5 w, g	FCR, 1–3 w, g/g	FCR, 3–5 w, g/g	FCR, 1–5 w, g/g	EPEI
Effect of rice bran										
0%	986 ^a^	1037 ^b^	2023	1279 ^a^	1720	2999	1.30	1.66 ^a^	1.48	424.1
5%	959 ^b^	1037 ^b^	1996	1248 ^b^	1713	2961	1.30	1.65 ^ab^	1.48	415.1
10%	910 ^c^	1067 ^a^	1977	1183 ^c^	1731	2914	1.29	1.62 ^b^	1.47	421.2
Effect of feed additives										
Liposorb	951 ^a^	1030 ^b^	1981	1221	1708 ^b^	2929	1.28	1.66	1.48	407.7 ^b^
Vit. E-Se	916 ^b^	1067 ^a^	1983	1198	1751 ^a^	2949	1.31	1.64	1.49	425.5 ^a^
Interaction rice bran x feed additive										
0%	986	1037	2023	1279	1720	2999	1.30	1.66	1.48	424.2
5%	961	1047	2008	1254	1724	2978	1.30	1.65	1.48	424.4
5% Liposorb	951	1026	1977	1231	1712	2943	1.30	1.67	1.49	396.0
5% Vit. E-Se	964	1038	2002	1259	1704	2963	1.31	1.64	1.48	425.1
10%	910	1068	1978	1200	1692	2892	1.32	1.58	1.46	418.3
10% Liposorb	951	1035	1986	1211	1705	2916	1.27	1.65	1.47	419.4
10%Vit. E-Se	868	1096	1964	1138	1798	2936	1.31	1.64	1.49	425.9
RMSE	17.5	14.9	0.423	26.6	18.6	27.4	0.021	0.026	0.016	13.2
*p*-values										
Rice bran	<0.0001	0.0001	0.0007	<0.0001	0.043	0.0001	0.551	0.004	0.221	0.511
Feed additive	0.003	0.0002	0.216	0.091	0.0001	0.333	0.029	0.011	0.184	0.034
Interaction	0.0001	0.009	0.010	0.003	<0.0001	0.066	0.204	0.079	0.049	0.090
Contrast *p*-value ^1^										
0 vs. 10% all RB groups	<0.0001	0.007	0.0005	0.0004	0.043	<0.0001	0.159	0.0005	0.073	0.536
5 vs. 10% all RB groups	<0.0001	0.0001	0.004	<0.0001	0.025	0.0005	0.946	0.015	0.189	0.274
0 vs. 5% RBNS	<0.0001	0.007	0.0005	0.0004	0.043	<0.0001	0.159	0.0005	0.073	0.536
0 vs. 10% RBNS	0.0005	0.060	0.01	0.01	0.025	0.0003	0.319	0.003	0.079	0.524
Liposorb vs. Vit. E-Se	0.294	0.257	0.024	0.159	0.562	0.325	0.483	0.147	0.425	0.005
5% vs. 5%RB Liposorb	0.399	0.057	0.007	0.240	0.373	0.089	0.547	0.227	0.589	0.006
10% vs. 10% RB Liposorb	0.004	0.005	0.553	0.592	0.326	0.243	0.005	0.002	0.574	0.912
10% vs. 10% RB Vit. E-Se	0.003	0.016	0.194	0.004	0.0001	0.035	0.581	0.006	0.009	0.423

^a,b,c^ Means in the same column with no common superscripts are significantly different (*p* < 0.05); RB—rice bran, Vit. E-Se—vitamin E+ selenium; RBNS—rice bran not supplemented; EPEI—European production efficiency index. W—week; RMSE—root mean square error. ^1^ Contrast value was presented only when signifcant <0.05 or approached signifcant <0.10.

**Table 4 vetsci-10-00299-t004:** Carcass traits and relative organs (%) of broiler chickens as affected by dietary rice bran treatments during the experiment.

	Carcass	Dressing	Heart	Liver	Gizzard	Giblets
Effect of rice bran						
0%	71.4 ^a^	77.5 ^a^	0.920	2.21 ^a^	2.97 ^a^	6.10 ^a^
5%	70.6 ^b^	76.3 ^b^	0.932	2.00 ^b^	2.86 ^b^	5.97 ^b^
10%	70.3 ^c^	75.9 ^c^	0.891	1.92 ^c^	2.75 ^c^	5.56 ^c^
Effect of feed additives						
Liposorb	70.5	76.5	0.905	1.97	2.84 ^a^	5.71 ^a^
Vit. E-Se	70.6	76.2	0.910	1.94	2.76 ^b^	5.60 ^b^
Interaction rice bran x feed additive						
0%	71.4	77.5	0.920	2.21	2.97	6.10
5%	70.3	76.0	0.928	2.00	2.89	5.82
5% Liposorb	70.7	76.5	0.908	1.99	2.87	5.77
5% Vit. E-Se	70.8	75.6	0.960	2.01	2.83	5.80
10%	70.3	75.9	0.910	1.96	2.75	5.62
10% Liposorb	70.2	75.9	0.903	1.95	2.81	5.66
10%Vit. E-Se	70.4	75.8	0.860	1.86	2.69	5.40
RMSE	0.147	0.204	0.039	0.092	0.063	0.076
*p*-value						
Rice bran	<0.0001	<0.0001	0.059	<0.0001	<0.0001	<.0001
Feed additive	0.002	0.079	0.778	0.204	0.039	0.009
Interaction	0.002	0.008	0.050	0.053	0.367	0.004
Contrast *p*-value ^1^						
0 vs. 5% all RB groups	<.00001	<0.0001	0.789	<0.0001	0.086	<0.0001
0 vs. 10% all RB groups	<0.0001	<0.0001	0.721	<0.0001	<0.0001	<0.0001
5 vs. 10% all RB groups	0.0006	<0.0001	0.019	0.002	0.0002	<0.0001
5 vs. 10% RBNS groups	<0.0001	<0.0001	0.789	<0.0001	0.086	<0.0001
0 vs. 5% RBNS	<0.0001	<0.0001	0.721	<0.0001	<0.0001	<0.0001
0 vs. 10% RBNS	0.348	0.611	0.534	0.352	0.005	0.002
5% vs. 5% RB Liposorb	0.0001	0.004	0.478	0.893	0.656	0.401
5% vs. 5% Vit. E-Se	<0.0001	0.001	0.253	0.687	0.173	0.789
10% vs. 10% RB Vit. E-Se	0.509	0.313	0.085	0.009	0.158	0.0007

^a,b,c^ Means in the same column with no common superscripts are significantly different (*p* < 0.05); RB—rice bran, Vit. E-Se—vitamin E+ selenium; RBNS—rice bran not supplemented; RMSE—root mean square error.^1^ Contrast value was presented only when signifcant <0.05 or approached signifcant <0.10.

**Table 5 vetsci-10-00299-t005:** Plasma protein components and liver and kidney functions in broiler chickens as affected by dietary rice bran treatments during the experiment.

	TP g/dL	Alb g/dL	Glb g/dL	Alb/Glb	ALT IU/L	AST IU/L	ALT/AST	Urea mg/dL	Crea mg/dL	Urea/Crea
Effect of rice bran										
0%	3.61 ^c^	3.14 ^a^	0.47 ^c^	6.68 ^a^	14.56	242.7	0.061	5.18	0.475	11.28
5%	3.89 ^b^	3.14 ^a^	0.75 ^b^	4.19 ^b^	13.33	244.8	0.054	4.88	0.417	11.26
10%	3.94 ^a^	2.62 ^b^	1.32 ^a^	1.98 ^c^	13.23	247.1	0.029	4.30	0.417	10.99
Effect of feed additives										
Liposorb	4.21 ^a^	3.24 ^a^	0.97 ^b^	3.34 ^a^	11.21	243.5	0.049	5.52	0.450	12.11
Vit. E-Se	3.56 ^b^	2.27 ^b^	1.29 ^a^	1.76 ^b^	13.27	239.5	0.056	3.94	0.388	10.58
Interaction effect between rice bran and feed additive										
0%	3.61	3.14	0.47	6.68	14.56	242.7	0.061	5.18	0.475	11.28
5%	3.70	3.10	0.60	5.17	16.17	259.4	0.062	5.08	0.425	11.55
5% Liposorb	3.92	3.19	0.73	4.37	12.11	235.2	0.051	6.18	0.400	14.52
5% Vit. E-Se	4.06	3.13	0.93	3.37	11.71	239.7	0.051	3.38	0.425	7.70
10%	4.27	3.18	1.09	2.92	14.57	250.1	0.058	3.53	0.400	9.83
10% Liposorb	4.49	3.28	1.21	2.71	10.29	251.8	0.041	4.88	0.500	9.70
10%Vit. E-Se	3.06	1.41	1.65	0.85	14.82	239.4	0.062	4.50	0.350	13.45
RMSE	0.062	0.073	0.082	0.185	6.66	16.1	0.027	0.154	0.073	2.96
*p*-value										
Rice bran	<0.0001	0.0001	<0.0001	<0.0001	0.123	0.459	0.185	0.446	0.394	0.926
Feed additive	<0.0001	0.032	<0.0001	<0.0001	0.471	0.174	0.562	0.125	0.251	0.523
Interaction	<0.0001	0.046	0.004	0.342	0.709	0.286	0.707	0.187	0.07	0.006
Contrast *p*-value ^1^										
0 vs. 5% all RB groups	0.0597	0.453	0.041	0.0002	0.055	0.158	0.091	0.928	0.345	0.897
0 vs. 10% all RB groups	<0.0001	0.110	<0.0001	<0.0001	0.104	0.523	0.139	0.147	0.162	0.497
5 vs. 10% all RB groups	<0.0001	<0.0001	<0.0001	<0.0001	0.970	0.721	0.922	0.374	0.999	0.828
5 vs. 10% RB NS groups	0.059	0.543	0.041	0.0002	0.055	0.159	0.091	0.928	0.345	0.897
0 vs. 5% RBNS	<0.0001	0.110	<0.0001	<0.0001	0.104	0.523	0.139	0.147	0.162	0.497
0 vs. 10% RBNS	<0.0001	0.024	<0.0001	0.0001	0.738	0.425	0.815	0.172	0.634	0.420
Liposorb vs. Vit. E-Se	0.0031	0.211	0.002	0.002	0.932	0.695	0.979	0.019	0.633	0.004
5% vs. 5%RB Liposorb	<0.0001	0.077	0.049	0.215	0.399	0.046	0.561	0.327	0.634	0.171
5% vs. 5% Vit. E-Se	<0.0001	0.572	<0.0001	0.0001	0.355	0.099	0.543	0.136	0.999	0.081
10% vs. 10% RB Liposorb	<0.0001	0.327	0.004	0.291	0.375	0.883	0.381	0.232	0.067	0.953
10% vs. 10% RB Vit. E-Se	<0.0001	0.003	<0.0001	0.010	0.958	0.361	0.825	0.384	0.345	0.098

^a,b,c^ Means in the same column with no common superscripts are significantly different (*p* < 0.05). ALT—alanine aminotransferase; AST—aspartate aminotransferase; RB—rice bran, Vit. E-Se—vitamin E+ selenium. RBNS—rice bran not supplemented; RMSE—root mean square error. ^1^ Contrast value was presented only when signifcant <0.05 or approached signifcant <0.10.

**Table 6 vetsci-10-00299-t006:** Blood lipid profile of broiler chickens as affected by dietary rice bran treatments during the experiment.

	Total Cholesterol, mg/dL	Triglycerides, mg/dL	LDL, mg/dL	HDL, mg/dL	LDL/HDL	vLDL, mg/dL
Effect of rice bran						
0%	193.8	184.8 ^ab^	109.4	47.4	2.31	37.0 ^ab^
5%	197.8	136.3 ^b^	121.8	48.9	2.50	27.3 ^b^
10%	215.5	217.3 ^a^	123.7	48.4	2.56	43.5 ^a^
Effect of feed additives						
Liposorb	202.6	181.4	118.9	47.1 ^b^	2.50	36.3
Vit. E-Se	213.8	197.2	124.3	50.1 ^a^	2.49	39.4
Interaction effect between rice bran and feed additive						
0%	193.8	184.8	109.4	47.4	2.31	36.9
5%	183.9	141.9	106.6	48.9	2.18	28.4
5% Liposorb	215.1	123.4	142.6	47.9	2.97	24.7
5% Vit. E-Se	194.9	143.6	116.3	49.9	2.35	28.7
10%	223.9	161.6	143.5	48.1	3.00	32.3
10% Liposorb	190.0	239.4	95.3	46.9	2.03	47.9
10%Vit. E-Se	232.6	250.7	132.3	50.2	2.63	50.2
RMSE	21.8	51.1	26.0	1.47	0.553	10.2
*p*-value						
Rice bran	0.138	0.003	0.617	0.342	0.682	0.003
Feed additive	0.548	0.222	0.878	0.006	0.920	0.223
Interaction	0.011	0.140	0.011	0.660	0.014	0.140
Contrast *p*-value ^1^						
0 vs. 10% all RB groups	0.067	0.529	0.079	0.539	0.089	0.527
5 vs. 10% all RB groups	0.065	0.0009	0.863	0.371	0.802	0.0009
0 vs. 5% RBNS	0.067	0.529	0.079	0.539	0.089	0.527
0 vs. 10% RBNS	0.018	0.593	0.059	0.423	0.048	0.593
Liposorb vs. Vit. E-Se	0.209	0.583	0.169	0.063	0.127	0.584
5% vs. 5% RB Liposorb	0.058	0.613	0.065	0.348	0.058	0.612
10% vs. 10% RB Liposorb	0.042	0.044	0.017	0.271	0.022	0.044
10% vs. 10% RB Vit. E-Se	0.581	0.023	0.543	0.054	0.352	0.023

^a,b^ Means in the same column with no common superscripts are significantly different (*p* < 0.05). HDL—high-density lipoprotein; LDL—low-density lipoprotein; VLDL—very low-density lipoprotein; RB—rice bran; Vit. E-Se—vitamin E+ selenium; RBNS—rice bran not supplemented; RMSE—root mean square error. ^1^ Contrast value was presented only when signifcant <0.05 or approached signifcant <0.10.

**Table 7 vetsci-10-00299-t007:** Antioxidant and immunological indices of broiler chickens as affected by dietary rice bran treatments during the experiment.

	Malondialdehyde (nmol/mL)	Glutathione Unit	Glutathione Peroxidase (ng/mL)	Superoxide Dismutase (U/mL)	IgA (ng/mL)	Ig M (mg/dl)	Ig G (mg/dl)
Effect of rice bran							
0%	97.3	0.320	22.3	22.3	769.3	640.7	2957
5%	81.6	0.443	26.8	30.4	530.1	439.2	2016
10%	106.9	0.365	21.3	23.3	804.1	667.6	2952
Effect of feed additives							
Liposorb	103.2	0.415	23.8	26.1	684.7	569.2	2560
Vit. E-Se	97.2	0.398	22.7	25.0	755.8	625.0	2783
Interaction effect between rice bran and feed additive							
0%	97.3	0.320	22.3	22.3	769.3	640.7	2957
5%	64.3	0.469	29.5	34.0	389.0	328.0	1574
5% Liposorb	103.6	0.354	23.0	25.0	657.7	542.0	2450
5% Vit. E-Se	76.7	0.506	28.0	32.3	543.7	447.7	2025
10%	100.3	0.329	21.8	24.4	732.6	604.2	2648
10% Liposorb	102.6	0.476	24.7	27.8	711.7	596.3	2670
10%Vit. E-Se	117.6	0.290	17.3	17.7	968.0	802.3	3540
RMSE	40.6	0.174	7.31	10.31	294.4	239.4	1114.1
*p*-value							
Rice bran	0.281	0.429	0.159	0.157	0.059	0.052	0.079
Feed additive	0.579	0.976	0.719	0.713	0.423	0.420	0.484
Interaction	0.538	0.151	0.235	0.243	0.432	0.444	0.509
Contrast *p*-value ^1^							
0 vs. 5% all RB groups	0.264	0.239	0.181	0.125	0.805	0.079	0.095
5 vs. 10% all RB groups	0.142	0.283	0.076	0.105	0.034	0.030	0.053
5 vs. 10% NS RB groups	0.264	0.239	0.181	0.125	0.083	0.079	0.095

RB—rice bran; Vit. E-Se—vitamin E+ selenium; RBNS—rice bran not supplemented; RMSE—root mean square error. ^1^ Contrast value was presented only when signifcant <0.05 or approached signifcant <0.10.

## Data Availability

The data are available upon official request from the principal investigator upon the permission of the funding agent.

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
