# Peer review of "Rice Bran as an Alternative Feedstuff in Broiler Nutrition and Impact of Liposorb® and Vitamin E-Se on Sustainability of Performance, Carcass Traits, Blood Biochemistry, and Antioxidant Indices"

_vetsci, 2023, doi:10.3390/vetsci10040299_

Round 1

Reviewer 1 Report

Dear authors, in the attachment there is a recommendation for correcting your manuscript.

Author Response

The paper is interesting, the topic scientifically popular, but necessary to make some corrections.

Au: thank you very much for your comment

Summary

Line 17-19 these sentences should be deleted from the Summary. They are more appropriate for

the part of Introduction.

Au: deleted

The summary should be rewritten, because it does not contain any results that the authors obtained

in the research.

Au: the abstract has been modified by including some results

Introduction

No clear aim of the research was written.

Au: the aim of the research has been clarified

Material and Methods

From the materials and methods, it is not clear whether you added the amounts of Vitamin E-Se and Liposorb® through the Premix, or whether you mixed it directly into the mixture for the chickens. Given that the Premix already contains vitamins E and Se, (as can be seen from the composition of the Premix below tables 1 and 2), the conclusion that the addition of these nutrients in the groups (3; 4; 6 i 7) is not correct, because the food had a higher content of the mentioned nutrients.

Groups:  The Vit E-Se and liposorb were add at the top of the diet replacing the same amount of feed mixture.

3) 5% rice bran +0.5 g/kg of Liposorb®,

4) 5% rice bran+1 g/kg of vitamin E-Se,

6) 10% rice bran + 0.5 g/kg Liposorb®,

7) 10%, rice bran + 1 g/kg of vitamin E-Se.

In the recommendation section we considered the supplemented amount Vit  E+Se regardless of the diet content as these Premix contents were regularly/usually added to the diet  

Discussion

Line 269-287 show the results of other authors. It is necessary to include the results of this research in the discussion. It is necessary to improve this part.

Au: thank you for your comment, we improved this section.

Conclusion

Line 308-309 considering that the table 4. showing the characteristics of the carcass only shows the yield and percentage of the carcass, and the percentage of offal (gizzard, heart, liver...) and not the proportions of basic parts of the carcass or the proportions of basic tissues in the most valuable parts of the carcass (breast and drumsticks ) the conclusion must be changed.

Au: changed

Reviewer 2 Report

Dear authors, 

Before a detailed revision of the paper, I have detected some limitations that impede the acceptance of your document in the current presentation:

1. For a nutritional study, the chemical composition provided is insufficient. You are evaluating a rice bran that could affect the content of fibre of the diet. Please, analyse your diets including at least DM, ashes, NDF, ADF, ADL and starch.

2. In the same way, in the studies where a raw material is analysed it is crucial to include the characterization of this raw material, especially with by-products. These by-product are quite variable, and it is very important to know the quality of the product used in this trial, including its complete chemical composition.

3. Performance traits must be analysed with a mixed model for repeated measures, and contrast of interest must be also included. Currently, the document includes the P-value of the treatment. It is necessary to include also the P-values of the level of rice bran, Liposorb and vitamin E-Se using contrast analysis.

If the authors make these changes, I will be happy to carry out a more detailed review of their work.

Author Response

Dear authors,

Before a detailed revision of the paper, I have detected some limitations that impede the acceptance of your document in the current presentation:

  1. For a nutritional study, the chemical composition provided is insufficient. You are evaluating a rice bran that could affect the content of fibre of the diet. Please, analyse your diets including at least DM, ashes, NDF, ADF, ADL and starch.

Au: Added. Thank you very much for your comment

  1. In the same way, in the studies where a raw material is analysed it is crucial to include the characterization of this raw material, especially with by-products. These by-product are quite variable, and it is very important to know the quality of the product used in this trial, including its complete chemical composition.

Au: Added. Thank you very much for your comment

  1. Performance traits must be analysed with a mixed model for repeated measures, and contrast of interest must be also included. Currently, the document includes the P-value of the treatment. It is necessary to include also the P-values of the level of rice bran, Liposorb and vitamin E-Se using contrast analysis.

Au: We believe that one way ANOVA would provide the best solution to this work regardless of the mean effects which may be confounded by the interaction effects when it significant.  The one way will gave better results about the effect of the additives within each level of rice bran and this is the final goal to accept/reject of our hypothesis. Thus, to avoid the confounded results of the interaction with the main effects we accepted one way and multicomparsion post hoc test with Tukey to avoid type 1 experimental error.

If the authors make these changes, I will be happy to carry out a more detailed review of their work. Thank you for valuable comments

Reviewer 3 Report

Dear Authors,

the manuscript deals with an important topic in this historical phase such as using of agricultural by-products in animal nutrition. Despite the not exactly positive results, the study is valid in the form and search for valid alternative food. The paper is well-written and considers aspects that can play a key role in the diffusion of new ingredients in animal nutrition.

Suggestions

Line

65 [13] please add Cosentino et al., 2020

Cosentino, C.; Freschi, P.; Fascetti, S.; Paolino, R.; Musto, M. Growth control of herbaceous ground cover and egg quality from an integrated poultry-hazelnut orchard system. Ita. J. Agronomy 202015, 3 214–221.

Line

97-99 “Liposorb® emulsifier was bought from Ceva Polchem Private Limited, Shivajinagar, Pune- 411016, India; each 1 kg contains 500g lecithin, 100 g polyethylglycerol ricinoleate, 2.16g sodium chloride, 9.0 g silicon dioxide, and 388.84 calcium carbonate.”

Please substituite the sentence with “Liposorb® emulsifier was bought (Ceva Polchem Private Limited, Shivajinagar, Pune- 411016, India), each 1 kg contains: 500 g lecithin, 100 g polyethylglycerol ricinoleate, 2.16 g sodium chloride, 9.0 g silicon dioxide, and 388.84 calcium carbonate.”

Line

126-128 “Collection of blood samples (n=7 per treatment, representing all treatment replicates) was accomplished at the end of the trail in heparinized tubes, after which they were cen-trifuged at rpm of 2,146 g for 15 minutes to separate the plasma sample.”

Please substituite the sentence with “At the end of trial, the collection of blood samples (n=7 per treatment, representing all treatment replicates) was accomplished in heparinized tubes, then they were centrifuged at rpm of 2,146 g for 15 minutes to separate the plasma sample.”

Line

171 “heart percentahe.”

Please substitute with “heart percentage.”

Line

190 Please revised the table 5:  the abscissa formatting is not aligned.

Line

199  Please revised the table 6:  the abscissa formatting is not aligned.

Line

209 Please revised the table 7.

Line

253 Please substituite with “-8.0%) in body weight gains of birds fed 30 and 40%”.

Line

331-424 Please revised the font of the references.

Author Response

Dear Authors,

the manuscript deals with an important topic in this historical phase such as using of agricultural by-products in animal nutrition. Despite the not exactly positive results, the study is valid in the form and search for valid alternative food. The paper is well-written and considers aspects that can play a key role in the diffusion of new ingredients in animal nutrition.

Au: thank you very much for your comment

65 [13] please add Cosentino et al., 2020

Cosentino, C.; Freschi, P.; Fascetti, S.; Paolino, R.; Musto, M. Growth control of herbaceous ground cover and egg quality from an integrated poultry-hazelnut orchard system. Ita. J. Agronomy 2020, 15, 3 214–221.

Au: done

97-99 “Liposorb® emulsifier was bought from Ceva Polchem Private Limited, Shivajinagar, Pune- 411016, India; each 1 kg contains 500g lecithin, 100 g polyethylglycerol ricinoleate, 2.16g sodium chloride, 9.0 g silicon dioxide, and 388.84 calcium carbonate.”

Please substituite the sentence with “Liposorb® emulsifier was bought (Ceva Polchem Private Limited, Shivajinagar, Pune- 411016, India), each 1 kg contains: 500 g lecithin, 100 g polyethylglycerol ricinoleate, 2.16 g sodium chloride, 9.0 g silicon dioxide, and 388.84 calcium carbonate.”

Au: the period has been corrected

126-128 “Collection of blood samples (n=7 per treatment, representing all treatment replicates) was accomplished at the end of the trail in heparinized tubes, after which they were cen-trifuged at rpm of 2,146 g for 15 minutes to separate the plasma sample.”

Please substituite the sentence with “At the end of trial, the collection of blood samples (n=7 per treatment, representing all treatment replicates) was accomplished in heparinized tubes, then they were centrifuged at rpm of 2,146 g for 15 minutes to separate the plasma sample.”

Au: the period has been corrected

171 “heart percentahe.”

Please substitute with “heart percentage.”

Au: done

190 Please revised the table 5:  the abscissa formatting is not aligned.

Au: corrected

199  Please revised the table 6:  the abscissa formatting is not aligned.

Au: corrected

209 Please revised the table 7.

Au: corrected

253 Please substituite with “-8.0%) in body weight gains of birds fed 30 and 40%”.

Au: corrected

331-424 Please revised the font of the references.

Au: corrected

Reviewer 4 Report

Dear Editor and Authors,

Despite the fact that the article was well-written, some essential points were overlooked. I have included a list of recommendations and suggestions for enhancing your work that I believe will be useful to you, as well as some shortcomings I have identified.

Major comments:

This study has a major concern about statistical analysis. The use of one-way ANOVA is inappropriate and may lead to type II errors. Ideally, we would use a factorial design (3 x 3), where two factors are considered, namely the level of rice bran (0, 5%, and 10%) and the additives (control, Liposorb, and Vitamin E-Se).

L103: It would be beneficial to combine Tables 1 and 2 into one. Furthermore, since one-week-old broiler chicks were used in the study, "starter-1-21 d" should be changed to "starter-8-21 d.

L116: Please clarify how feed consumption was determined at the end of the first week for chicks purchased for one week of age.

L121-122: “7 birds from each treatment” or “one bird from each replication” ? It is also possible to select all of them from one group. Please check this statement.

L163-164: The body weight of birds in the third week should also be reported in Table 3.

The discussion needs to be enhanced with real explanations, not only agreements, disagreements and similarities. The authors should improve the discussion by demonstrating the biological basis for the achieved results. If possible, please discuss the potential mechanisms behind the observations you have made.

Minor comments:

L34: Please use “metabolic” instead of “metaboilc”.

L43: It would be better to replace "very interesting" with "of great interest".

L46, L61, L242, L283: Please replace “rations” with “diet” in order to be consistent with the rest of the text.

L66: Please use “.” İnstead of “,” before Abou-Kassem et al. [13].

L69, L71: Please use “Liposorb® and Vitamin E-Se” instead of “Vitamin E-Se and Liposorb®” (For consistency with the rest of the text).

L73-75: It would be helpful if this sentence was linked to the previous one.

L136-139: This sentence does not contain a verb. Please check it.

L156: It is recommended that RB be abbreviated where it is first used.

L158: Please remove “…or without…”.

L182: Please use “of” İnstead of “od”.

L183, L188: Please use “(P <0.001)” İnstead of “(P <0.01)”.

L191: Please use  “Albumin/globulin ratio” and “<0.001” instead of  “Albumin/golbulin ratio” and “0.0001”.

Author Response

Dear Editor and Authors,

Despite the fact that the article was well-written, some essential points were overlooked. I have included a list of recommendations and suggestions for enhancing your work that I believe will be useful to you, as well as some shortcomings I have identified.

Au: thank you for your comment

Major comments:

This study has a major concern about statistical analysis. The use of one-way ANOVA is inappropriate and may lead to type II errors. Ideally, we would use a factorial design (3 x 3), where two factors are considered, namely the level of rice bran (0, 5%, and 10%) and the additives (control, Liposorb, and Vitamin E-Se).

Au We believe that one way ANOVA would provide the best solution to this work regardless of the mean effects which may be confounded by the interaction effects when it significant. The one way will gave better results about the effect of the additives within each level of rice bran and this is the final goal to accept/reject of our hypothesis. Thus, to avoid the confounded results of the interaction with the main effects we accepted one way and multicomparsion post hoc test with Tukey to avoid type 1 experimental error.

L103: It would be beneficial to combine Tables 1 and 2 into one. Furthermore, since one-week-old broiler chicks were used in the study, "starter-1-21 d" should be changed to "starter-8-21 d.

Au: done

L116: Please clarify how feed consumption was determined at the end of the first week for chicks purchased for one week of age.

Au: We started the experiment at the age of one week and did not determine any parameters before this period. 

L121-122: “7 birds from each treatment” or “one bird from each replication” ? It is also possible to select all of them from one group. Please check this statement.

Au: the statement is correct: 1 bird from each replication. In a group we have 7 replications, so 7 birds for each treatment.

L163-164: The body weight of birds in the third week should also be reported in Table 3.

Au: Done as required - thank you for your comment

The discussion needs to be enhanced with real explanations, not only agreements, disagreements and similarities. The authors should improve the discussion by demonstrating the biological basis for the achieved results. If possible, please discuss the potential mechanisms behind the observations you have made.

Au: Some improvements are included in the discussion

Minor comments:

L34: Please use “metabolic” instead of “metaboilc”.

Au: done

L43: It would be better to replace "very interesting" with "of great interest".

Au: done

L46, L61, L242, L283: Please replace “rations” with “diet” in order to be consistent with the rest of the text.

Au: done

L66: Please use “.” İnstead of “,” before Abou-Kassem et al. [13].

Au: done

L69, L71: Please use “Liposorb® and Vitamin E-Se” instead of “Vitamin E-Se and Liposorb®” (For consistency with the rest of the text).

Au: done

L73-75: It would be helpful if this sentence was linked to the previous one.

Au: done

L136-139: This sentence does not contain a verb. Please check it.

Au: corrected

L156: It is recommended that RB be abbreviated where it is first used.

Au: done

L158: Please remove “…or without…”.

Au: done

L182: Please use “of” İnstead of “od”.

Au: done

L183, L188: Please use “(P <0.001)” İnstead of “(P <0.01)”.

Au: done

L191: Please use “Albumin/globulin ratio” and “<0.001” instead of  “Albumin/golbulin ratio” and “0.0001”.

Au: done

Round 2

Reviewer 1 Report

The proposed corrections have been accepted in the paper, and the paper can be printed in this form in the journal.

Author Response

Thank you very much.

Reviewer 2 Report

Dear authors,

Although you believe that an ANOVA analysis is appropriate for the statistical analysis of the performance traits, this is not right. When you have repeated measures along the time for the same variable if you applied a one-way ANOVA each time you include the effect of "the animal within treatment" to the treatment effect, this is not appropriate and leads to erroneous results. For this reason,  these data must be reanalysed using a mixed model for repeated measures. 

In addition, to evaluate separately the effect of the rice bran level, Liposorb and vitamin E-Se it is needed the use of contrast analysis. I'm not requesting you to include the effect of them in the model, you can use a one-way model for the variable "treatment", but to determine the effect of each of them separately, you must include contrast of interest to obtain the P-values of each factor included in your treatment variable. This is key to ensure that the three factors evaluated have an effect itself. 

Author Response

The statistical analysis has been changed, by adding contrast analysis, too. As a consequence, the "Statistical analysis" section and the tables from 2 to the last have been modified. The discussion has been partly modified according to the new tables.

Reviewer 4 Report

Dear Editor and Authors,

The manuscript has been significantly improved from the first draft. Nonetheless, my objections in the statistical analysis part that I highlighted in my previous assessment report remain relevant for the amended work. It is anticipated that there would be a change in various parameters in a research utilizing 0, 5%, and 10% rice bran due to the dosage effect. I think you need to modify the statistical model in order to suggest adding Liposorb® or Vitamin E-Se to a diet including rice bran at any rate up to 10% if there is no dosage impact, as you claimed in your research.

Best regards

Author Response

(The authors gave the same response as above.)
